# Application of Sawdust-Derived Activated Carbon as a Bio-Based Filler in Vulcanized Rubber Bushings

**DOI:** 10.3390/polym17222996

**Published:** 2025-11-11

**Authors:** Enasty Pratiwi Wulandari, Popy Marlina, Heryoki Yohanes, Wahju Eko Widodo, S. Joni Munarso, Eko Bhakti Susetyo, Yenni Bakhtiar, Haixin Guo, Wahyu Bahari Setianto

**Affiliations:** 1Mechanical Engineering Department, Tamansiswa University, Palembang 30126, Indonesia; enastypratiwi@gmail.com; 2The Palembang Center for Standardization and Industrial Services (BSPJI), Palembang 30128, Indonesia; popymarlina16@gmail.com; 3National Research and Innovation Agency (BRIN), KST B.J. Habibie, South Tangerang 15314, Indonesia; lanj001@brin.go.id (L.); hery009@brin.go.id (H.Y.); wahj003@brin.go.id (W.E.W.); sjon001@brin.go.id (S.J.M.); astu001@brin.go.id (A.); ekob002@brin.go.id (E.B.S.); yenn001@brin.go.id (Y.B.); wahy013@brin.go.id (W.B.S.); 4Agro-Environmental Protection Institute, Ministry of Agriculture and Rural Affairs, Tianjin 300191, China; haixin_g@126.com

**Keywords:** activated carbon, bio-based filler, mechanical properties, sawdust-derived activated carbon, vulcanized rubber bushings

## Abstract

This study investigated sawdust-derived activated carbon (SAC) as a sustainable reinforcing filler for vulcanized rubber bushings (VRBs). Two types SAC200 (75 µm, carbonized at 200 °C) and SAC400 (38 µm, carbonized at 400 °C) were chemically activated and incorporated into natural rubber (NR) at 25–55 phr loadings, while SAC free VRBs served as controls. Fourier transform infrared (FTIR) analysis revealed that SAC400 exhibited stronger hydroxyl and carbonyl functional groups, indicating higher surface reactivity compared with SAC200. The incorporation of SAC increased cross-linking density, thereby enhancing both curing behavior and mechanical performance. VRBs reinforced with SAC400 demonstrated higher maximum torque (up to 38.07 kg·cm), shorter scorch time (5 min 58 s), and reduced cure time (11 min 05 s) relative to SAC200 and the control. Mechanical properties improved markedly, with hardness and tensile strength rising from 45 Shore A and 5.52 MPa in the control to 70 Shore A and 13.40 MPa in SAC400. Although elongation at break decreased slightly, it remained within the acceptable range for dynamic applications. Swelling resistance also increased, reaching 101.76% at 25 °C and 106.61% at 100 °C. Overall, SAC400 consistently outperformed SAC200 and the control, highlighting its potential as a renewable, biomass-derived filler for high-performance rubber bushings and promising a sustainable alternative to conventional fillers in industrial applications.

## 1. Introduction

Natural rubber (NR) is an elastomeric biopolymer composed of cis-1,4-polyisoprene [1,2]. It is one of the most widely used elastomers, with global production reaching 13.77 million tons in 2021 [3]. NR is extensively applied across diverse sectors, including automotive components, medical devices, petrochemicals, mining, and other industrial applications [4]. The growing diversity of NR-based applications has stimulated the development of advanced compounding systems, which incorporate various additives such as accelerators, activators, plasticizers [5], antioxidants [6], vulcanizing agents, and reinforcing fillers. Among these, reinforcing fillers play a pivotal role in enhancing the mechanical and dynamic performance of vulcanizates, including tensile strength, modulus, elongation at break, abrasion resistance, compression set, swelling resistance, and dimensional stability [7,8,9]. Nearly all commercial rubber vulcanizates utilize reinforcing fillers derived from petroleum-based carbon black (CB) [10,11]. Petroleum-derived CB is produced via incomplete combustion of hydrocarbons, where approximately two metric tons of petroleum feedstock are required to produce one metric ton of CB [12]. However, this reliance on petroleum-based CB imposes significant environmental burdens.

Among these, reinforcing fillers play a pivotal role in enhancing the mechanical and dynamic performance of vulcanizates, including tensile strength, modulus, elongation at break, abrasion resistance, compression set, swelling resistance, and dimensional stability [12,13,14,15]. In automotive applications, rubber bushings are essential for vibration isolation, enabling controlled movement between components while reducing noise and improving both comfort and safety [13,14,15]. To optimize these properties, conventional fillers such as CB are widely employed. Nevertheless, concerns regarding environmental impact and the depletion of fossil resources have driven interest in sustainable biomass-derived alternatives [16,17]. Bio-based fillers (bio-fillers) offer several advantages, including biodegradability, a lower environmental footprint, reduced density, and cost-effectiveness [18]. Their compatibility with rubber matrices further strengthens their potential for developing green composites [16].

The mechanical behavior of rubber bushings is strongly influenced by polymer type, filler loading, and compounding formulation [19]. Hence, the integration of bio-fillers into NR not only aligns with sustainable materials engineering but also provides promising mechanical reinforcement. Several bio-derived fillers have been investigated in NR systems. Cifriadi et al. [12] reported the utilization of cassava starch and tamarind wood, while Haslilywaty et al. [20] employed rice husk ash and tacca starch. Joshi et al. [21] demonstrated the reinforcing effects of banana fibers. These bio-fillers improved mechanical performance while also offering biodegradability [22]. Thomas et al. [23] highlighted the advantageous properties of cellulose-, chitin-, and starch-based fillers, which are cost-effective and environmentally friendly. Furthermore, Sowińska et al. and Suhu [24] reported improved thermal oxidation resistance and reinforcement using walnut shell powder and corn starch, respectively. 

Activated carbon derived from lignocellulosic biomass, particularly sawdust, has recently emerged as a promising bio-filler due to its high surface area, porous structure, and abundant surface functionalities [25,26]. According to Negara et al., [27] sawdust-derived activated carbon exhibits mesoporous structures with an average pore diameter of 2.43 nm, a pore volume of 0.369 cm^3^ g^−1^, and a specific surface area 409.7 m^2^ g^−1^, enabling broad applicability. These properties facilitate both mechanical interlocking and physicochemical interactions with rubber matrices. Although its reinforcing potential is recognized, systematic studies remain limited on how processing parameters, particularly carbonization temperature and particle size affect its performance.

Rubber bushings operating under dynamic and cyclic loading conditions require high compressive strength, elasticity, and fatigue resistance [28]. While conventional fillers such as CB and silica provide excellent reinforcement [29,30], they often increase stiffness at the expense of elasticity. Consequently, bio-based fillers are sought as alternative reinforcements that can maintain mechanical performance while offering environmental benefits. Activated carbon from renewable precursors is an inexpensive and abundant alternative with the potential to generate added value [31].

Sawdust-derived activated carbon (SAC) typically exhibits high surface area, porosity, and functional group density, which promote strong interactions with polymer matrices. For example, SAC has been reported to possess a hierarchical porous structure with micropores (0.55 nm) and mesopores (2.58 nm), resulting in a specific surface area of 621 m^2^ g^−1^ and a pore volume of 0.35 cm^3^ g^−1^ [32]. With such characteristics, SAC enhances interfacial interactions within rubber matrices. Previous studies synthesized SAC via chemical activation followed by carbonization 43. Chemically activated SAC exhibited favorable properties, including a BET surface area of ~1250 m^2^ g^−1^, a pore volume of 0.78 cm^3^ g^−1^, carbon content of ~93%, low ash content (7.36%), and an iodine number of 1161.3 mg g^−1^ [33]. Incorporation of SAC into rubber compounds at loadings up to 100 phr improved torque response and hardness.

Despite these advantages, most prior studies employed only a single SAC variant, limiting insights into how carbonization temperature and particle size govern reinforcement efficiency. Understanding these parameters is critical for tailoring filler morphology and functionality toward optimal performance in NR vulcanizates, particularly for high-demand automotive components such as bushings. This study aims to investigate the influence of carbonization temperature (200 °C and 400 °C) and particle size (75 µm and 38 µm) of SAC on the curing characteristics and mechanical properties of NR composites for automotive bushing applications. SAC200 and SAC400 were incorporated at various loadings and compared with an unfilled control. Fourier transform infrared spectroscopy (FTIR) was employed to explore surface chemistry variations, while vulcanization behavior (maximum torque, scorch time, and optimum cure time) and mechanical properties (tensile strength, elongation at break, hardness, and compression set) were systematically evaluated.

This work contributes to the development of environmentally friendly, high-performance rubber composites while supporting biomass waste valorization in advanced materials engineering. The incorporation of SAC into NR-based bushing compounds enhances interfacial polar interactions absent in conventional CB, resulting in superior reinforcement efficiency. Moreover, the findings emphasize that SAC surface chemistry plays a decisive role in determining the mechanical properties of rubber bushings.

Unlike previous studies, this study uses a single variant of activated carbon derived from sawdust. This study systematically explores the combined effects of carbonization temperature and particle size on the performance of reinforcing fillers in rubber vulcanizates. The novelty of this study lies in demonstrating how these processing parameters govern the drying behavior and mechanical response of rubber bushings under dynamic conditions. The results provide new insights into the tailoring of biomass-derived fillers for high-performance elastomer applications, while also promoting sustainable materials engineering through the utilization of sawdust waste.

## 2. Methods

### 2.1. Instrumentation

The preparation of the rubber compounds and vulcanized samples was conducted using the equipment listed in Table 1. Key instruments included a muffle furnace (Nabertherm GmbH) for sawdust carboniation, an XK-160 open two-roll mill for mastication and compounding, and a hydraulic hot press for vulcanization. Additional equipment included a digital balance for accurate weighing of materials and a Monsanto Moving Die Rheometer (MDR 2000, Alpha Technologies, Bellingham, WA, USA) for measuring vulcanization characteristics.

### 2.2. Materials and Specifications

The materials utilized in this experiment are listed in Table 2.

### 2.3. Experimental Design

#### 2.3.1. Preparation of Activated Charcoal

Sawdust was first washed with distilled water and oven-dried at 110 °C for 24 h. The carbonization process was conducted in a muffle furnace at 900 °C for 30 min to obtain raw charcoal, which was then ground and sieved to 200 mesh (75 µm) and 400 mesh (38 µm). The chemical activation process was initiated by soaking 5000 g of sieved charcoal in a 10% phosphoric acid (H_3_PO_4_) solution for 24 h at ambient temperature. Following impregnation, the samples were thoroughly rinsed with distilled water to remove residual acid. Subsequent carbonization was performed in a muffle furnace equipped with a gas circulation system utilizing hydrogen (H_2_) and nitrogen (N_2_), along with a hydrogen combustion safety unit. Two types of sawdust-derived activated charcoal were produced: SAC200—particle size of 75 µm, carbonized at 200 °C for 120 min; SAC400—particle size of 38 µm, carbonized at 400 °C for 120 min.

Post-carbonization, the activated charcoal was allowed to cool within the furnace for 24 h to prevent oxidation and then stored in airtight containers to maintain its integrity. The equipment utilized in this process is shown in Figure 1.

The quality parameters of the resulting activated charcoal were evaluated in accordance with SNI 06-3730-1995 standards (refer to Table 3) [34]. 

#### 2.3.2. Rubber Compounding Procedure

Rubber compounding was conducted based on the formulation presented in Table 4 using a two-roll open mill. Natural rubber was first masticated for approximately 8 min until a plasticized state was achieved. The activator (ZnO) and co-activator (stearic acid) were then added and mixed for 3 min to ensure uniform dispersion. SAC was gradually incorporated and milled for 4 min until evenly distributed within the rubber matrix. Subsequently, the plasticizer, antioxidant, accelerator, scorch retarder, and coupling agent were sequentially introduced and mixed for 5 min to achieve a fully homogeneous compound. Finally, the vulcanizing agent (sulfur) was added and mixed for an additional 5 min until the compound reached complete uniformity.

#### 2.3.3. Vulcanization

The rubber compounds were molded and vulcanized using a hydraulic hot press at 150 °C for 10 min under a pressure of 170 kgf/cm^2^ to produce vulcanized rubber bushing (VRB) specimens.

### 2.4. Test Method

#### 2.4.1. FTIR Spectroscopy

Fourier transform infrared (FTIR) spectroscopy was performed in this study using a Shimadzu IRTracer-100 spectrometer equipped with an Attenuated Total Reflectance (ATR) accessory to identify the surface functional groups present on the sawdust-derived activated carbon (SAC). Special attention was given to the detection of hydroxyl (–OH) and carbonyl (C=O) groups, which are known to promote interfacial interactions with elastomeric matrices and enhance filler–rubber compatibility. The presence of these oxygen-containing groups plays a crucial role in improving filler dispersion, interfacial adhesion, and overall reinforcement efficiency in rubber composites [35].

#### 2.4.2. Curing Characteristics

Curing behavior was evaluated using a Moving Die Rheometer (MDR 2000) at 150 °C, following ISO 3417:2008 [36]. The parameters measured included minimum torque (Smin), maximum torque (Smax), scorch time (ts_2_), and optimum cure time (t_90_) [37,38]. The cure rate index (CRI) [39] was calculated using the following equation:(1)$$CRI=\;\frac{100}{{tc}_{90\;-\;t_{2\;}}}\;\times \;100\%$$

#### 2.4.3. Mechanical Testing Procedures

Mechanical properties were evaluated through standard test methods. Hardness was measured using a durometer in accordance with ASTM D2240-15 [40]. Tensile strength according to ASTM D395-18 (22 h at 100 °C) [40]. Swelling resistance was assessed following ASTM D471 [41] by immersing samples in engine oil at 100 °C for 72 h. The percentage swelling [42] was calculated using the following equation:(2)$$\%\;Swelling=\;\frac{W_{2}-\;W_{1}}{W_{1}}\;\times \;100\%\;$$
where W_1_ is the mass of the specimen prior to immersion, and W_2_ is the mass following immersion.

All mechanical tests were conducted in triplicate, and the results are presented as the mean ± standard deviation.

## 3. Results and Discussion

### 3.1. FTIR Analysis of Sawdust-Based Activated Carbon

The FTIR spectrum of sawdust-derived activated carbon (SAC) (Figure 2) revealed the presence of characteristic functional groups that contribute to the reinforcing capability of SAC in the natural rubber matrix. A broad absorption band centered at 3639–3138 cm^−1^ corresponds to O–H stretching vibrations from hydroxyl groups in phenolic, alcoholic, or adsorbed moisture species.

These polar functionalities increase surface polarity, enabling hydrogen bonding and improved filler–rubber interactions. The bands observed at 2609.77 cm^−1^ and 1720–1842 cm^−1^ are assigned to C=O stretching vibrations, suggesting the presence of carbonyl groups from carboxylic acids, aldehydes, or ketones generated during oxidative activation. Such oxygenated groups are known to enhance interfacial reactivity and promote higher crosslinking density during vulcanization [36]. In addition, weak peaks near 2920 cm^−1^ (C–H stretching of –CH_2_/–CH_3_), 1481.38 cm^−1^ (C–H bending), and 1109.11 cm^−1^ (C–O stretching of secondary alcohols or ethers) further confirm the partial retention of lignocellulosic structures and the introduction of oxygen-based moieties. The fingerprint region (500–1100 cm^−1^) also exhibits bands characteristic of alcohols, ethers, and esters, while the peak at 686.68 cm^−1^ corresponds to aromatic C=C stretching, indicating the presence of condensed aromatic structures derived from lignin carbonization [43]. While FTIR provides direct evidence of surface functional groups, the confirmation of elemental composition and chemical states requires complementary analysis. X-ray Photoelectron Spectroscopy (XPS) has been widely recommended in recent studies to quantify oxygen-containing groups (C–O, C=O, O–C=O) and to verify their chemical environment. For instance, XPS deconvolution of the C 1s region typically reveals sp^2^-hybridized carbon (C=C), sp^3^-hybridized carbon (C–C), and oxygenated species (C–O, C=O), in agreement with the oxygen functionalities detected by FTIR. The O 1s spectrum further distinguishes between hydroxyl, carbonyl, and carboxyl contributions. Therefore, the combination of FTIR and XPS provides a robust verification: FTIR identifies vibrational modes of functional groups, while XPS quantifies the atomic-level chemical states and confirms the abundance of oxygen functionalities that facilitate strong interfacial adhesion in SAC-filled rubber composites. The FTIR–XPS results in this study are in line with the report from Rosan et al. [44], Yang et al. [45], Marija Ilić et al. [46], where FTIR detects the characteristic vibrations of oxygenated groups (C–O, C=O), while XPS quantifies the contribution of oxygen species through the deconvolution of C1s and O1s peaks. This consistency strengthens the evidence that the presence of oxygen groups plays an important role in enhancing interfacial interactions in activated carbon-based materials.

This dual characterization not only validates the surface chemistry of SAC but also underpins its reinforcing role in natural rubber compounds formulated in Table 4. The simultaneous presence of hydroxyl, carbonyl, ether, and aromatic structures contributes to improved filler dispersion, increased cross-link density during vulcanization, and enhanced mechanical performance of vulcanized rubber bushings compared to conventional carbon black. Such complementary evidence highlights SAC as a sustainable and multifunctional bio-based filler with clear advantages in polymer reinforcement applications [47,48].

### 3.2. Curing Characteristics of Natural Rubber Compounds

The curing characteristics of natural rubber (NR) compounds filled with SAC200 and SAC400 at various loadings (0–55 phr) were evaluated using rheometric parameters, including maximum torque (S_max_), minimum torque (S_min_), torque difference (Δ_S_ = S_max_ − S_min_), scorch time (ts_2_), optimum cure time (t_90_), and cure rate index (CRI) [49]. The results, summarized in Table 5, reveal significant variations in vulcanization behavior across formulations. The chemical functionality of SAC, particularly the presence of oxygen-containing groups like hydroxyl and carbonyl, which improve interactions with curing agents and the rubber matrix, is primarily responsible for these variations [50]. FTIR test results indicate the presence of dominant hydroxyl and carbonyl groups in SAC400. The presence of these groups affects the hardening characteristics of the rubber compound. With the presence of this group, the maximum torque (Smax) of SAC400 vulcanizate increased significantly from 3.451 N·m to 3.733 N·m when compared with SAC200 and the control.

The difference in filler loading has a significant effect on cross-link formation. At a filler loading of 30 phr, the cross-link density increases and the torque delta (Δ_S_) changes. This implies that the network structure and stiffness of the rubber have improved [51]. However, at filler loadings above 30 phr, Δ_S_ decreases and the ts_2_ and t_90_ values increase. This phenomenon implies that excessive amounts of filler can cause the rubber chain to become unstable and the resulting cross-link system to become less uniform.

Among the measured parameters, maximum torque (S_max_) rose significantly with increasing SAC content, particularly for SAC400. A S_max_ of 38.07 kg·cm was recorded by SAC400 at 55 phr, indicating increased stiffness and cross-link density. This was ascribed to enhanced filler–matrix adhesion, which was facilitated by the polar surface and numerous active sites of SAC400 [52,53,54,55]. Conversely, minimum torque (S_min_), which reflects compound viscosity before curing, showed only minor variation with SAC loading. Yet, SAC400 showed a marked increase in Δ_S_ at higher loadings, peaking at 29.31 kg·cm at 45 phr, likely due to its finer particle size (38 µm), which enhances surface area and dispersion efficiency [56]. SAC400, with a surface area of 38 µm, has a higher density of polar functional groups (–OH, –COOH, –C=O). These groups act as catalytic sites that can interact with sulfur during vulcanization. These groups accelerate the cross-linking reaction. Furthermore, vulcanizates loaded with SAC400 exhibited a faster ts_2_ time than SAC200, resulting in a faster cross-linking reaction.

Similar trends have been reported in previous studies, where activated carbon with abundant surface functionalities accelerated the drying kinetics and enhanced the rubber–filler interactions. Li et al. [57] showed that activated carbon with very high surface area had abundant oxygen functionalities, while Ma et al. [58] reported that Sakura-derived activated carbon exhibited significant –OH, –C=O, and –COOH groups. The results indicate that the bio-based activated carbon filler accelerated the drying time and improved the mechanical performance of natural rubber composites, consistent with the faster ts_2_ observed in the SAC400-filled vulcanizate.

The SAC material used in this study possessed high porosity and carbon content (72.32%), contributing to better surface reactivity and adsorption capacity [59]. These properties enabled more efficient heat transfer and molecular mobility during vulcanization, leading to denser crosslinking and improved mechanical response. Consequently, both scorch time (ts_2_) and optimum cure time (t_90_) decreased with SAC400 loading, reaching their lowest values at 5 min 58 sec and 11 min 05 sec, respectively. Acceleration of the curing process, in line with previous findings that finer particles increase thermal conductivity and internal friction, thereby shortening the overall curing time [60]. A shorter t_90_ increases energy efficiency in processing [61,62], while a lower ts_2_ improves the processability of the compound.

The cure rate index (CRI), representing vulcanization kinetics, was highest for SAC400 at 55 phr, confirming its superior curing performance [63]. SAC400’s enhanced surface properties high surface area, porosity, and polarity are key contributors to this outcome, promoting faster heat distribution and chain mobility. As observed in Table 5, the trend of increasing Smax with SAC loading suggests a corresponding increase in cross-link density, likely driven by interactions between SAC hydroxyl groups and the NR matrix [64]. Meanwhile, S_min_ values fluctuated moderately, likely influenced by the balance between filler dispersion and matrix viscosity.

At equivalent loadings, SAC400 consistently outperformed SAC200 in terms of ts_2_, t_90_, and CRI, confirming its superior curing efficiency. Variations in cross-link density and internal heat generation largely governed the differences in curing behavior from filler–rubber friction [65]. Overall, NR compounds filled with 45–55 phr SAC400 exhibited the most favorable curing characteristics, high Smax, low ts_2_, t_90_, and elevated CRI values. These findings position SAC400 not only as a reinforcing filler but also as an active contributor to vulcanization kinetics. Its dual role underscores its potential as an efficient, bio-based alternative to conventional fillers, particularly for demanding applications such as automotive rubber bushings [66].

### 3.3. Mechanical Properties and Performance Analysis

#### 3.3.1. Hardness

Hardness is a key mechanical parameter reflecting the resistance of a vulcanizate to surface deformation and is directly related to cross-linking density and filler–matrix interactions. In automotive rubber bushings (VRBs), which must maintain dimensional stability under repeated mechanical stress, hardness serves as a crucial performance metric. As shown in Figure 3 and Table 4, the addition of SAC to natural rubber compounds can significantly increase hardness.

The unfilled control sample exhibited a baseline hardness of 45 Shore A. With increasing SAC loading, a progressive rise in hardness was observed. SAC400, carbonized at 400 °C and possessing a finer particle size (38 µm), consistently yielded higher hardness values than SAC200 (75 µm, 200 °C). At 55 phr, SAC400 reached 70 Shore A, fulfilling the industrial standard for VRB (70 ± 5 Shore A). Because SAC200 could only support 60 Shore A at the same loading, the reinforcement was less effective. Activated carbon with smaller particle size and richer surface functionality provides better reinforcement in the rubber matrix, resulting in higher hardness values compared to larger particle fillers.

The superior performance of SAC400 is attributed to its higher surface area and carbon purity, which promote effective filler dispersion and interfacial bonding. FTIR analysis (Figure 2) confirmed the presence of functional groups such as C=C and C=O, along with reduced –OH intensity in SAC400, indicating greater graphitization and hydrophobicity. The detection of sulfur-containing bonds (C–S, S–S) further indicates that SAC400 can participate in the vulcanization reaction, contribute to the formation of three-dimensional networks, and influence the VRB hardness value. The enhancement in hardness is closely related to increased crosslink density. The observed increase in hardness is primarily due to the presence of functional groups in SAC, as identified through FTIR analysis. These functional groups enhance interfacial interactions with the rubber matrix. Furthermore, cross-linking formed by sulfur, accelerated by accelerators and activators, increases the overall cross-linking density. This contributes to the increase in VRB hardness. These findings are consistent with previous findings on bio-based fillers [67]. The use of the ZnO stearic acid system, in combination with CBS and TMTD accelerators, supported efficient vulcanization and uniform crosslink distribution. However, hardness plateaued at loadings above 45 phr, likely due to filler agglomeration, which may hinder matrix uniformity and restrict further reinforcement.

The presence of coupling agents during mixing reduces interfacial tension and improves filler–rubber adhesion [68]. In modified natural rubber (MNR), polar functional groups can interact favorably with SAC, while non-polar domains align with the rubber matrix. Larger SAC particles may limit chain mobility during deformation, resulting in greater localized indentation and decreased hardness [69]. Overall, SAC400-filled NR compounds at 45 and 55 phr met commercial hardness requirements, demonstrating the material’s capability for application in high-performance automotive bushings (VRB). The combination of optimized morphology and surface chemistry supports SAC400 as a viable bio-derived alternative to conventional fillers in elastomeric systems.

#### 3.3.2. Tensile Strength

Tensile strength is a fundamental parameter for assessing the mechanical performance of vulcanized rubber (VRB), especially in reinforced systems. It represents the material’s ability to withstand stress under tensile loading before failure. Figure 4 presents the tensile strength of VRB filled with SAC200 and SAC400 across various loadings. 

The unfilled compound exhibited a tensile strength of 5.52 MPa, establishing the baseline for comparison. As SAC content increased, tensile strength improved progressively, confirming SAC’s reinforcing capability. The increased tensile strength is due to improved filler–matrix adhesion, interfacial stress transfer, and physical interlock, all of which are facilitated by SAC’s porous structure and large surface area. These results are consistent with earlier studies on the enhancement of elastomeric properties by carbon fillers derived from biomass [70]. The existence of polar functional groups on the filler surface explains the higher tensile strength of VRB, particularly those reinforced with SAC400.

The polar groups actively encourage greater interfacial adhesion by forming hydrogen bonds and dipole–dipole interactions with the non-polar polyisoprene chains. Interfacial adhesion improves stress transfer effectiveness in the composite. Furthermore, the polar surface properties of SAC400 promote better dispersion into the NR matrix. The resulting interfacial adhesion interacts with ZnO, stearic acid, sulfur, and other additives to form tighter cross-links. This results in a significant increase in tensile strength.

Across all loadings, SAC400 outperformed SAC200. At 45 phr, SAC400 reached a peak tensile strength of 13.4 MPa, significantly higher than SAC200’s 9.56 MPa at the same loading, an increase of approximately 40%. This value exceeds the tensile strength range typically reported for carbon black-filled NR (11–13 MPa at 30 phr) [71,72], highlighting SAC400’s competitive performance. The superior performance of SAC400 is attributed to its finer particle size (38 µm), higher carbonization temperature (400 °C), and higher surface area, which collectively enhance dispersion and interfacial bonding. At 55 phr, SAC400 maintained a high tensile strength of 12.51 MPa, surpassing the minimum industry requirement of 11 MPa for automotive VRB. In contrast, SAC200 exhibited a marked reduction at this loading, likely due to agglomeration and poor dispersion, common challenges in particulate-filled elastomer systems.

The reinforcing mechanism involves both physical and chemical interactions. Physically, SAC particles interact with rubber chains via van der Waals forces. Chemically, functional groups on the SAC surface (–OH, C=O) form bonds with the rubber matrix during vulcanization [72]. The presence of maleated natural rubber (MNR) as a coupling agent further enhances compatibility and stress distribution. SAC400’s optimized morphology also contributes to a more uniform crosslink network, resulting in higher tensile strength [73]. These results support earlier findings on the importance of filler particle size, surface area, and dispersion in achieving mechanical reinforcement [74]. Overall, SAC400 demonstrated superior performance compared to SAC200, positioning it as a highly effective, sustainable filler for high-performance vulcanized rubber bushings.

#### 3.3.3. Elongation at Break

Elongation at break is an important parameter that reflects the flexibility and toughness of VRB under tensile stress. This parameter indicates the extent to which a material can stretch before breaking. As shown in Figure 5, the unfilled control sample (0 phr SAC) exhibited an elongation at break of 530%, demonstrating unrestricted chain mobility in the absence of fillers. The SAC200 and SAC400 loadings on each VRB significantly affected these properties. The loading of SAC200 and SAC400 on each VRB significantly affected these properties, consistent with previous findings that the addition of bio- or carbon-based fillers limits chain mobility and reduces elongation at break due to stronger filler–rubber interactions. According to Azura et al. [71], elongation at break is influenced by filler load, where increasing filler load will reduce elongation at break.

The gradual addition of SAC fillers reduced elongation due to restricted polymer chain mobility, as the rigid particles acted as physical crosslinking points. In SAC200-filled compounds, elongation values declined notably, especially at 35–55 phr, where values fell below the minimum industrial standard of 400%. This was attributed to poor dispersion, weak filler–rubber interfacial bonding, and filler agglomeration, which disrupted matrix homogeneity and increased stiffness.

At higher SAC content, the reduction in free volume further hindered chain flexibility. Excess filler led to saturation, preventing complete interaction between the SAC and the VRB. This condition compromises elasticity and promotes premature failure under tensile loading. In contrast, SAC400 demonstrated superior elongation-at-break performance. At 25 phr and 35 phr, elongation at break reached 435% and 460%, respectively, within the optimal range (400–500%) for VRB applications. However, further increasing the loading to 45 phr and 55 phr reduced elongation to 415% and 370%, respectively. This reduction was due to filler aggregation, which restricted chain extension and introduced stress concentration points.

The improved performance of SAC400 is largely due to its smaller particle size (38 µm) and higher carbonization temperature (400 °C), which results in greater surface area and porosity. These characteristics enhance stress distribution across the VRB, interfacial adhesion, and filler dispersion. SAC400’s higher porosity resulting from thermal activation also allowed for tighter rubber filler contact and better stress transfer, as shown in similar studies using biomass-based carbon [54]. By comparison, SAC200 with a larger particle size (75 µm) led to uneven stress distribution, localized microvoids, and poor elongation results at higher loadings. These phenomena underline the importance of filler structure and dispersion quality in mechanical reinforcement [75].

The 35 phr SAC400 formulation represented the optimal filler level, where elongation at break peaked at 460%. Above this threshold, the performance declined below the 400% standard, reflecting the onset of filler overloading. Similar behavior was observed in carbon-filled composites post-thermal treatment, as reported [76]. Notably, SAC-filled VRB outperformed conventional carbon black systems, which typically exhibit maximum elongation between 350% and 400% at 20–30 phr [77]. In comparison, SAC400 reached 460% at 35 phr, demonstrating its superior reinforcement efficiency. This also highlights SAC400’s viability as a sustainable alternative to petroleum-based fillers, combining mechanical reliability with environmental advantages, wide availability, and renewability. When considered together, the data confirm that SAC400 enhances both elasticity and toughness of VRB more effectively than SAC200, particularly at optimal loading. Its favorable elongation performance, alongside its bio-based origin, supports its use in the development of environmentally sustainable, high-performance rubber composites.

#### 3.3.4. Compression Set

Compression set is an important parameter for evaluating the long-term dimensional stability and elastic recovery of rubber materials, especially in VRB applications. Compression set measures the ability of a rubber vulcanizate to return to its original shape after prolonged compression at elevated temperatures [78]. Lower VRB compression set values indicate superior elasticity, structural resistance, and the material’s capacity to maintain mechanical integrity under thermal and compression loads [79]. Figure 6 presents the compression set values of VRB reinforced with various loadings of SAC200 and SAC400. All samples were aged at 100 °C for 72 h to simulate thermal degradation conditions. The control (unfilled) sample exhibited a compression set of 29.5%, typical of non-reinforced vulcanizates. In contrast, the addition of SAC fillers significantly reduced compression set values across all formulations, suggesting enhanced elastic recovery due to increased crosslink density and stronger filler–matrix interactions. Figure 6 presents the improved VRB compression set values with various SAC200 and SAC400 loadings.

For samples loaded with SAC200, the VRB compression set decreased from 24.2% at 10 phr to 20.3% at 30 phr, indicating improved dimensional stability. SAC400 consistently outperformed SAC200 at equivalent loads. At 30 phr, SAC400 achieved the lowest compression set value of 18.7%, representing a 36.6% reduction compared to the control. At 45 phr, SAC400 maintained a value around 24%, still lower than the SAC200 counterpart. 

These improvements are attributed to SAC400’s smaller particle size (38 µm), higher surface area, and more porous structure—resulting from carbonization at 400 °C, which facilitated better dispersion and stronger interfacial bonding with the rubber matrix [66,71]. The superior performance of SAC400 is further supported by its higher surface polarity and structural stability, which helped distribute applied pressure more evenly and restricted polymer chain relaxation [80]. This led to reduced permanent deformation and improved dimensional resilience. The enhanced filler–rubber bonding in SAC400 composites is due to greater contact area and a more favorable distribution of functional groups, consequences of its higher carbonization temperature and optimized microstructure.

A sharp decline in compression set was observed when filler loading exceeded 25 phr, suggesting the formation of a percolation network. This interconnected filler structure enhanced matrix reinforcement and minimized voids through increased physical contact and interfacial bonding [81]. Similar trends have been reported in elastomers filled with high-surface-area carbon materials, where enhanced stress transfer and limited chain mobility reduced permanent deformation [29]. These observations align with the study by Yan et al. [55], which demonstrated that finer carbon fillers significantly improve stress distribution and limit molecular relaxation. The 25 phr threshold appeared critical, beyond which compression set performance improved markedly [82]. Notably, SAC400-filled VRB in the 35–55 phr range showed compression set values between 24% and 30%, meeting the industrial requirement (≤30%) for automotive and general-purpose bushing applications.

The excellent compression set resistance of SAC400 is also linked to its high porosity and functional surface chemistry, which promote physical entrapment of polymer chains and the formation of van der Waals and hydrogen bonds [83]. These interactions increase network compactness, reduce voids, and enhance the rubber’s ability to recover from compressive strain. The synergy between small particle size, high carbonization temperature, and broad surface functionality highlights the potential of SAC400 as a sustainable active filler alternative derived from biomass.

#### 3.3.5. Swelling Behavior

Swelling behavior in VRB is primarily governed by free volume, crosslink density, and interfacial interactions between the rubber matrix and the swelling medium. The swelling rate, defined as the percentage of absorbed oil, reflects the extent of solvent diffusion within the polymer network [42]. In this study, swelling tests were conducted on VRB using oil immersion at two temperatures (25 °C and 100 °C), as shown in Figure 7.

The formulations are detailed in Table 4. At 25 phr filler loading, SAC200 and SAC400 exhibited the highest swelling ratios, indicating lower crosslink densities. SAC200 swelled up to 50% at 25 °C, while the unfilled compound showed ~55% swelling at 100 °C. This high swelling [84], is associated with insufficient crosslink formation, which permits solvent penetration and polymer chain expansion. Additionally, polar groups result in less swelling in non-polar solvents, which suggests improved network integrity and solvent resistance.

When the filler content increased to 55 phr, the swelling percentage of VRB decreased significantly. This phenomenon indicates increased cross-linking and improved compatibility of the filler with the rubber matrix due to the addition of SAC. VRB filled with SAC400 consistently showed lower swelling compared to SAC200. Furthermore, the denser three-dimensional network formed during vulcanization reduces solvent uptake by physically restricting chain mobility and void space. This effect is supported by FTIR results and aligns with previous findings that higher cross-linking density reduces matrix permeability and swelling [62]. Furthermore, the results of FTIR analysis showed that the presence of polar groups increased tissue integrity and solvent resistance by reducing the amount of swelling in non-polar solvents.

Swelling resistance was consistently lower at higher SAC loadings, particularly in the range of 35–55 phr for SAC400. These results emphasize the role of filler morphology and surface chemistry in limiting chain relaxation and solvent transport. This behavior is more pronounced at high temperatures (100 °C), where solvent diffusion is accelerated and mechanical integrity is reduced. Nevertheless, SAC-filled samples maintained better swelling resistance compared to unfilled VRB. Furthermore, strong interfacial bonding through hydrogen bonding, van der Waals forces, and chemical linkages contributed to improved matrix cohesion and solvent exclusion, This was especially evident in SAC400 systems, which offered superior dimensional and chemical stability under thermal and chemical stress. In summary, VRB filled with SAC400 exhibited superior swelling resistance compared to SAC200. This was attributed to the finer structure of SAC400, higher surface reactivity, and better integration into the rubber matrix. These findings support the use of SAC400 as a reinforcing filler to enhance cross-linking and reduce solvent-induced degradation of VRB in vulcanized rubber systems.

## 4. Conclusions

(1)Sawdust-derived activated carbon (SAC) successfully incorporates oxygenated and aromatic functional groups, as confirmed by FTIR and XPS.(2)The addition of SAC improves the hardening properties, tensile strength, modulus, hardness, and tear resistance of NR vulcanizates.(3)SAC200 retains more oxygenated groups, enhancing interfacial interactions, while SAC400 increases stiffness due to the increase in aromatic domains.(4)Compared with conventional carbon black and other biomass-derived fillers, SAC exhibits superior sustainability and reinforcement potential.

## Figures and Tables

**Figure 1 polymers-17-02996-f001:**
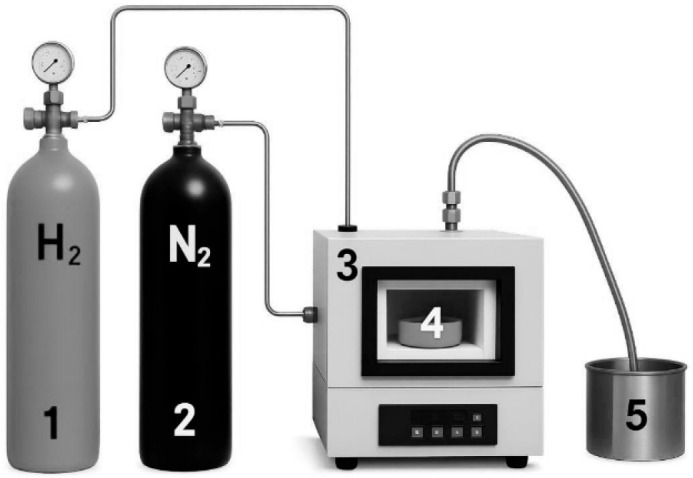
Schematic diagram of the carbonization equipment setup: (1) hydrogen gas cylinder, (2) nitrogen gas cylinder, (3) furnace, (4) carbonization reactor, and (5) nitrogen gas trap.

**Figure 2 polymers-17-02996-f002:**
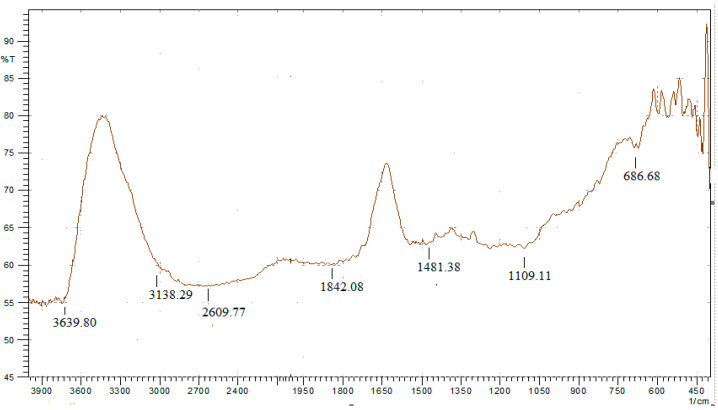
The FTIR spectrum of carbon black derived from sawdust indicates the presence of major functional groups.

**Figure 3 polymers-17-02996-f003:**
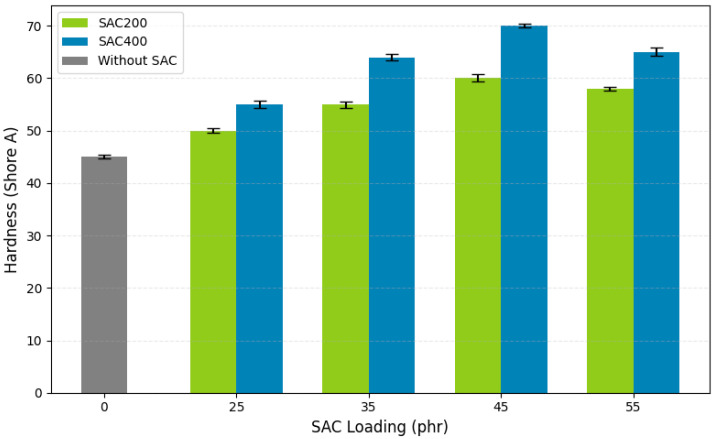
The effect of SAC type and loading on the Shore A hardness of vulcanized natural rubber bushings.

**Figure 4 polymers-17-02996-f004:**
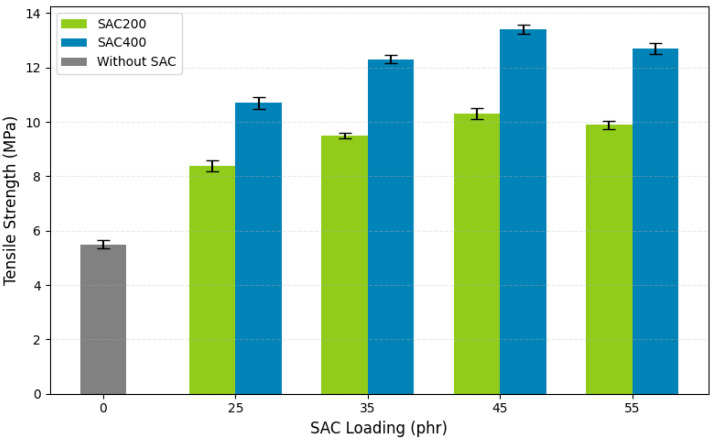
The effect of SAC type and loading on the tensile strength of vulcanized rubber bushings.

**Figure 5 polymers-17-02996-f005:**
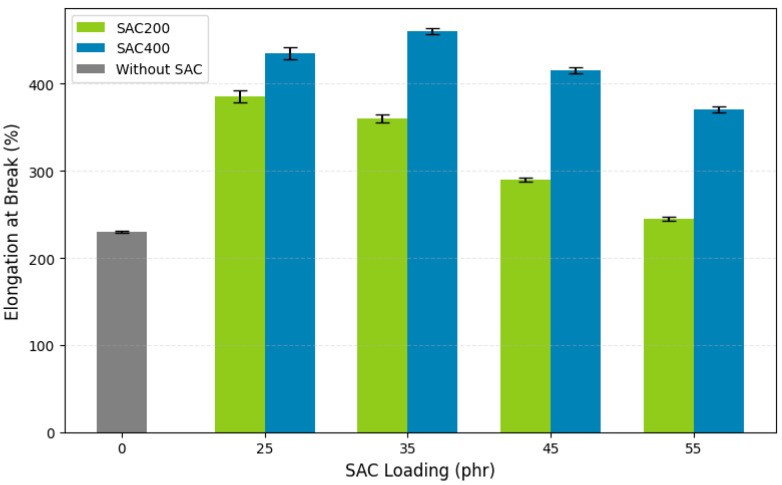
The effect of SAC type and loading on the elongation at break of vulcanized rubber bushings.

**Figure 6 polymers-17-02996-f006:**
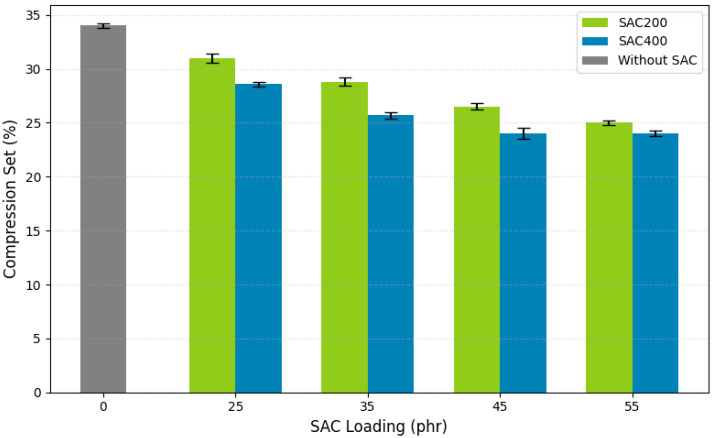
The effect of SAC type and loading on the compression set values of vulcanized rubber bushings.

**Figure 7 polymers-17-02996-f007:**
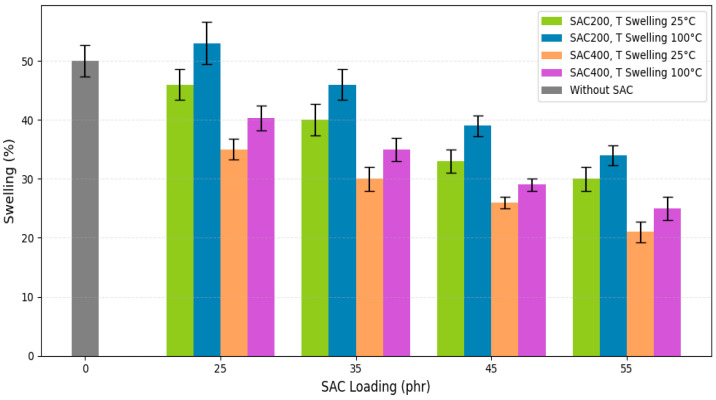
The effect of SAC type and loading on the oil swelling behavior of vulcanized rubber bushings.

**Table 1 polymers-17-02996-t001:** Equipment specifications for the preparation of rubber compounds.

No	Equipment	Specifications
1	Muffle Furnace (Nabertherm GmbH, Lilienthal, Germany)	Max Temp: 1700 °C; Volume: 0.15 ft^3^; Heating Elements: MoSi_2_; Controller: Shimaden FP93
2	XK-160 Open Mill (Qingdao Ouli Machine Co., Ltd., Qingdao, China)	Roll Ø: 160 mm; Working length: 320 mm; Motor: 7.5 kW
3	Hydraulic Press (HTM10042T, Toyo Seiki Seisaku-sho, Ltd., Tokyo, Japan)	Pressure: 100 Ton; Platens: 2 levels; Temp: up to 300 °C
4	Digital Scale (Ohaus Corporation, Parsippany, NJ, USA)	Capacity: 5 kg; Resolution: 0.1 g; LCD display

**Table 2 polymers-17-02996-t002:** Materials used in the formulation of rubber composites.

No	Chemical Name	Source and Specifications
1	Natural Rubber (SIR-20)	PT. Prasidha Aneka Niaga Tbk, Indonesia; Compliant with SNI 1903:2017 (Ash: 0.25%, Dirt: 0.042%, Nitrogen: 0.24%, Volatile matter: 0.2%, PRI: 65%, Mooney viscosity: 70)
2	SAC200 (75 µm), SAC400 (38 µm)	Local supplier, Palembang, Indonesia
3	Zinc oxide (ZnO)	Shanghai, China; CAS No.: 1314-13-2; Purity: 99.7%
4	Stearic acid	Guangdong, China; CAS No.: 57-11-4; Purity: 99%
5	TMTD	Jiangsu, China; CAS No.: 137-26-8; Purity: 98%
6	CBS	Fujian, China; CAS No.: 95-33-2; Purity: 96%
7	TMQ	Hunan, China; CAS No.: 793-47-8; Purity: 99.9%
8	Paraffin oil (PO-60)	PT. Pertamina, Indonesia
9	6PPD	China; CAS No.: 793-24-8; Purity: 96%
10	PVI	Henan, China; CAS No.: 17796-82-6; Purity: 99%
11	Maleated Natural Rubber (MNR)	Local supplier
12	Phosphoric Acid (H_3_PO_4_)	Zhejiang, China; Purity: 85%
13	Sulfur	Tamil Nadu, India; CAS No.: 7704-34-9; Purity: 99.5%
14	Engine Oil 10W/40 4T	Local supplier, Indonesia

Note: All materials were procured from local Indonesian vendors.

**Table 3 polymers-17-02996-t003:** Physicochemical characteristics of sawdust-based activated charcoal.

Parameter	Content
Water content	6.24%
Ash content	7.36%
Volatile matter	11.34%
Carbon	87.85%
Absorption of iodine	1161.34 mg/g

**Table 4 polymers-17-02996-t004:** Formulations of natural rubber compounds with varying SAC particle sizes and loadings (phr).

Material	Formula (Per Hundred Rubber, phr)								
	1	2	3	4	5	6	7	8	9
SIR 20	100	100	100	100	100	100	100	100	100
ZnO	5	5	5	5	5	5	5	5	5
Stearic acid	1	1	1	1	1	1	1	1	1
6PPD	2	2	2	2	2	2	2	2	2
TMQ	1	1	1	1	1	1	1	1	1
SAC200	-	25	35	45	55	-	-	-	-
SAC400	-	-	-	-	-	25	35	45	55
Minarex oil	1	1	1	1	1	1	1	1	1
CBS	3	3	3	3	3	3	3	3	3
TMTD	2	2	2	2	2	2	2	2	2
MNR	1	1	1	1	1	1	1	1	1
Sulfur	2	2	2	2	2	2	2	2	2
PVI	0.5	0.5	0.5	0.5	0.5	0.5	0.5	0.5	0.5

**Table 5 polymers-17-02996-t005:** Curing characteristics of natural rubber composites filled with SAC200 and SAC400.

Sample	Loading (phr)	Curing Characteristic					
		S_max_ (N·m)	S_min_ (N·m)	S_max_ − S_min_ (N·m)	Opt Cure Time (tc_90_) (min; s)	Scorch Time (ts_2_) (min; s)	CRI (s^−1^)
NR/CB Without filler	0	3.683	0.874	2.810	12:35	6:79	0.316
Without filler	0	3.451	0.796	2.655	15:59	8:45	0.230
SAC200 (75 µm)	25	3.037	0.610	2.427	15:29	8:02	0.223
	35	3.065	0.584	2.480	14:45	7:56	0.244
	45	3.143	0.512	2.631	13:57	7:25	0.255
	55	3.247	0.708	2.539	13:07	7:10	0.273
Without filler	0	3.451	0.796	2.655	15:59	8:45	0.230
SAC400 (38 µm)	25	3.241	0.775	2.466	14:57	7:47	0.233
	35	3.526	0.834	2.693	13:15	7:00	0.267
	45	3.722	0.847	2.874	12:28	6:46	0.292
	55	3.733	0.884	2.850	11:05	5:58	0.326

Note: Torque values originally expressed in kg·cm were converted to N·m using the conversion factor 1 kg·cm = 0.0980665 N·m. NR: natural rubber; phr: per hundred rubber; CB: carbon black; SAC: sawdust-activated carbon.

## Data Availability

The original contributions presented in this study are included in the article material. Further inquiries can be directed to the corresponding author.

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
