# Peer review of "Application of Sawdust-Derived Activated Carbon as a Bio-Based Filler in Vulcanized Rubber Bushings"

_polymers, 2025, doi:10.3390/polym17222996_

Round 1
Reviewer 1 Report
Comments and Suggestions for Authors
- The manuscript presents serious conceptual and scientific problems. The authors propose the development of a rubber compound reinforced with activated carbon derived from biomass, which could potentially be used to replace carbon black:
Line 48: To optimize these properties, conventional fillers such as carbon black are widely used. However, concerns about environmental impacts and the depletion of fossil fuels have driven interest in sustainable biomass-based alternatives [9,18]. Bio-fillers offer several advantages, including biodegradability, a lower environmental footprint, and reduced density and cost [19].
However, the reference material used in this study does not contain carbon black in its composition. How can the authors propose that activated carbon could replace carbon black if they do not present a reference containing carbon black? - In the abstract, the authors state that the FTIR results indicate that the SAC400 sample exhibits stronger hydroxyl and carbonyl groups:
Line 19: Fourier transform infrared (FTIR) analysis revealed that SAC400 exhibited stronger hydroxyl and carbonyl functional groups, indicating higher surface reactivity compared with SAC200.
First, the authors present only the spectrum of one sample, without clarifying which sample it refers to. How can they claim that SAC400 has advantages if they do not present both spectra? - The authors make unfounded statements based on the FTIR spectrum:
a. A peak around 2920 cm⁻¹ corresponds to C–H stretching of aliphatic –CH₂ and –CH₃ groups, suggesting partial retention of lignocellulosic structures post-activation and confirming incomplete degradation of the original biomass.
There is no peak at 2920 cm⁻¹ in the spectrum. - A strong band near 1720 cm⁻¹ is assigned to C=O stretching vibrations of carboxylic acids, ketones, or aldehydes formed during oxidative activation, enhancing SAC’s surface reactivity and interaction with vulcanized rubber [57].
There is no strong band at 1720 cm⁻¹ in the spectrum. - A prominent peak near 620 cm⁻¹, which corresponds to C=C stretching in aromatic rings, confirms the presence of condensed aromatic structures that result from the thermal breakdown of lignin [60].
It is not possible to use the band at 620 cm⁻¹ to confirm lignin thermal degradation. Undegraded lignin is already rich in aromatic rings. - The authors use references inappropriately:
The chemical functionality of SAC, particularly the presence of oxygen-containing groups like hydroxyl and carbonyl, which improve interactions with curing agents and the rubber matrix, is primarily responsible for these variations [62].
Reference 62 refers to a review on the presence of functional groups in supercapacitors. There is no relation to rubber and activated carbon. - The authors base much of the mechanical discussion on the premise of a possible reaction between functional groups during rubber vulcanization. However, the presence of these functional groups was not demonstrated, nor were any possible reaction mechanisms presented.
Author Response
Name : Nasruddin
Affiliation : National Research and Innovation Agency, Republic of Indonesia
email address : nasr006@brin.go.id
Date : 09-21-2025
To the Editor, Editor, Polymers (MDPI)
Subject: Submission of Revised Manuscript (ID: polymers-3867779)
Dear Editor and Reviewers
On behalf of all my co-authors, I am pleased to submit a revised version of our manuscript entitled “Application of Activated Carbon from Sawdust as a Bio-Based Filler in Vulcanized Rubber Bushings.”
We sincerely thank the editor and reviewers for their insightful and constructive comments on our work.
All suggestions have been carefully addressed, and this manuscript has been substantially improved in terms of clarity, accuracy, and overall scientific quality.
Detailed responses and revisions to each comment are available in the attached document "Responses to Reviewers."
All modifications are marked in the revised manuscript using color-coded Track Changes for easy evaluation. At the end of this appendix, we also present revisions from all reviewers in a single manuscript.
We believe this revised version now meets the journal's high standards and makes a clear and precise contribution to the development of sustainable bio-based fillers for rubber composites. We ask that you consider this revised manuscript for publication in Polymers.
Thank you very much for your time and consideration.
Sincerely,
Nasruddin
Corresponding Author

Reviewer 2 Report
Comments and Suggestions for Authors
The overall format of this manuscript is standardized, the logic is rigorous, the experimental design is scientific, and the result analysis is thorough, making it of great academic value. It is suggested to further revise and improve in accordance with the following opinions.
- The introduction is well-written at the end and has some general citations at the beginning. For instance, the first paragraph cites 13 references. It is recommended to select representative ones and introduce them appropriately, especially some quantitative data, to enhance the rigor of the manuscript.
- Regarding the verification of functional groups, it is recommended that on the basis of characterization by FTIR, the elemental composition and chemical state be analyzed by XPS to further prove the existence of relevant functional groups, and the two mutually confirm each other.
- In the discussion of the results, in addition to comparing with the control group, it is recommended to compare with the research achievements of other researchers in the literature to further illustrate the necessity and advancement of this study.
- In the conclusion, it is recommended to summarize and describe in points to facilitate reading and understanding.
- Relevant units should be international units as much as possible. For example, the torque unit should be N·m instead of kg·cm.
Author Response

(The authors gave the same response as above.)

Reviewer 3 Report
Comments and Suggestions for Authors
This is a well-written and comprehensive study on an important and timely topic: the development of sustainable, bio-based fillers for rubber composites. The manuscript is well-structured, the research is conducted with scientific rigor, and the overall findings represent a valuable contribution to the field. I recommend its publication after addressing some minor inconsistencies to further improve the manuscript’s clarity and accuracy.
I have identified several points that I recommend addressing:
- Incorrect ASTM Standard Citation: In Section 2.4.3 "Mechanical Properties", the standard for tensile strength is cited as ASTM D395-18. However, ASTM D395 is the standard for compression set. The correct standard for tensile testing is typically ASTM D412. Please verify and correct this citation.
- Contradiction in Elongation at Break Data: In the discussion for Section 3.3.3 "Elongation at Break", the text states that the unfilled control sample had an elongation at break of 530%. This value directly contradicts the data presented in Figure 5, where the "Without SAC" bar is approximately 230%. Please reconcile the text with the figure data.
- Contradiction and Incorrect Data Points in Compression Set Discussion: In Section 3.3.4 "Compression Set", the text claims the control sample had a compression set of 29.5% , which is inconsistent with Figure 6, where the value is approximately 34%. More importantly, this section references data for "10 phr" and "30 phr" loadings. These concentrations were not part of this study's experimental design as detailed in Table 4. Please remove these non-existent data points and correct the control value in the text.
- Major Discrepancy in Swelling Data: There is a significant discrepancy between the swelling data reported in the text and the corresponding figure. The Abstract and Conclusion state the lowest swelling values for SAC400 were 101.76% and 106.61%. However, Figure 7 shows that the maximum swelling percentage for any sample was below 60%. These values are fundamentally inconsistent. Please correct the values in the text to reflect the actual data shown in the figure.
- Minor Figure Citation Error: In the discussion for Section 3.3.1 "Hardness", the text refers to "Figure 3 and Table 4" to support the statement on hardness increase. Table 4 is the formulation table and does not contain hardness data. The reference to Table 4 should be removed.
Addressing these points will ensure the manuscript's data and discussion are fully aligned, which will significantly strengthen the credibility of this valuable research. I look forward to seeing the revised version.
Author Response

(The authors gave the same response as above.)

Round 2
Reviewer 1 Report
Comments and Suggestions for Authors
The authors have addressed all the points mentioned in the first revision.
Author Response
Thank you for your recognition.